# A Small Molecule Targeting the Intracellular Tyrosine Kinase Domain of ROR1 (KAN0441571C) Induced Significant Apoptosis of Non-Small Cell Lung Cancer (NSCLC) Cells

**DOI:** 10.3390/pharmaceutics15041148

**Published:** 2023-04-05

**Authors:** Amineh Ghaderi, Mohammad-Ali Okhovat, Jemina Lehto, Luigi De Petris, Ehsan Manouchehri Doulabi, Parviz Kokhaei, Wen Zhong, Georgios Z. Rassidakis, Elias Drakos, Ali Moshfegh, Johan Schultz, Thomas Olin, Anders Österborg, Håkan Mellstedt, Mohammad Hojjat-Farsangi

**Affiliations:** 1Department of Oncology-Pathology, BioClinicum, Karolinska University Hospital Solna, Karolinska Institutet, 171 64 Stockholm, Sweden; 2Kancera AB, Nanna Svartz Väg 4, 171 65 Solna, Sweden; 3Thoracic Oncology Center, Karolinska Comprehensive Cancer Center, 171 76 Solna, Sweden; 4Department of Immunology, Arak University of Medical Sciences, Arak 3848170001, Iran; 5Department of Pathology, Medical School, University of Crete, 700 13 Heraklion, Greece; 6Department of Hematology, Karolinska University Hospital Solna, 171 64 Solna, Sweden

**Keywords:** ROR1, NSCLC, small molecules, KAN0441571C, erlotinib, ibrutinib, TTP

## Abstract

The ROR1 receptor tyrosine kinase is expressed in embryonic tissues but is absent in normal adult tissues. ROR1 is of importance in oncogenesis and is overexpressed in several cancers, such as NSCLC. In this study, we evaluated ROR1 expression in NSCLC patients (N = 287) and the cytotoxic effects of a small molecule ROR1 inhibitor (KAN0441571C) in NSCLC cell lines. ROR1 expression in tumor cells was more frequent in non-squamous (87%) than in squamous (57%) carcinomas patients, while 21% of neuroendocrine tumors expressed ROR1 (*p* = 0.0001). A significantly higher proportion of p53 negative patients in the ROR1^+^ group than in the p53 positive non-squamous NSCLC patients (*p* = 0.03) was noted. KAN0441571C dephosphorylated ROR1 and induced apoptosis (Annexin V/PI) in a time- and dose-dependent manner in five ROR1^+^ NSCLC cell lines and was superior compared to erlotinib (EGFR inhibitor). Apoptosis was confirmed by the downregulation of MCL-1 and BCL-2, as well as PARP and caspase 3 cleavage. The non-canonical Wnt pathway was involved. The combination of KAN0441571C and erlotinib showed a synergistic apoptotic effect. KAN0441571C also inhibited proliferative (cell cycle analyses, colony formation assay) and migratory (scratch wound healing assay) functions. Targeting NSCLC cells by a combination of ROR1 and EGFR inhibitors may represent a novel promising approach for the treatment of NSCLC patients.

## 1. Introduction

Lung cancer is one of the main causes of death in cancer patients [1], and non-small cell lung cancer (NSCLC) accounts for around 85% of lung cancer cases. Current treatments for NSCLC are not sufficiently effective and there is an urgent need for new precision medicines [2].

Receptor tyrosine kinases (RTKs) are phosphotransferase enzymes that are essential for intracellular signal transduction and that are important targets for cancer treatment [3]. ROR1 (receptor tyrosine kinase-like orphan receptor 1), also known as neurotrophic tyrosine kinase receptor-related 1 (NTRKR1), is a transmembrane tyrosine-kinase enzyme encoded by a gene on chromosome 1. ROR1 belongs to the ROR family (ROR1 and ROR2) and was discovered based on amino acid sequence homology to the Trk family of neurotrophin receptors [4]. ROR1 is highly expressed during embryonic development in central neurons, and respiratory, cardiac, and skeletal tissues, and is essential for the proliferation, differentiation, polarity, and migration of neurons [5]. ROR1 is downregulated after birth but maintains a low expression in a few normal tissues, such as adipocytes, stomach and duodenum, pancreatic islets, parathyroid glands, and early B-cells [6,7,8]. Wnt5a is a ligand for ROR receptor proteins [9]. 

ROR1 is, however, expressed in hematologic malignancies and solid tumors, such as chronic lymphocytic leukemia (CLL) [10,11], mantle cell lymphoma (MCL) [12], diffuse large B-cell lymphoma (DLBCL) [13,14,15], melanoma [16,17,18,19,20,21,22], and NSCLC [17,23]. ROR1 is essential for the expansion, survival, epithelial-to-mesenchymal transition (EMT), migration, and metastasis of malignant cells [15,16,24,25,26]. ROR1 is involved in sustaining the self-renewal of cancer stem cells, and is related to disease activity and resistance to chemotherapy [27,28,29]. Several key signaling pathways such as PI3K/AKT/mTOR and the canonical/non-canonical Wnt pathways, as well as transcription factors such as CREB and C-Jun, have been proposed to be associated with ROR1 activation [30,31,32,33,34,35]. In various tumors, such as CLL, breast, and gastric cancers, high expression of ROR1 was associated with advanced disease and short survival [31,36,37,38,39,40].

ROR1 targeting agents, such as siRNA, monoclonal antibodies (mAbs), chimeric antigen receptor-modified T cells (CAR-T), and small molecule inhibitors (SMI), induced significant tumor cell death in hematological malignancies and cancers of epithelial origin [31,36,37,41,42,43,44,45]. The first-in-class ROR1 inhibitor (KAN0439834), which binds to the intracellular tyrosine kinase (TK) domain of ROR1, induced apoptosis of CLL cells and inhibited tumor cell growth in NOD/SCID mice xenotransplanted with human CLL cells [46]. KAN0439834 dephosphorylated ROR1 and inhibited the PI3K/AKT/mTOR pathway [20]. The combination of the ROR1 inhibitor with ibrutinib or erlotinib showed additive effects on pancreatic tumor cell death, compared to treatment with either agent alone [20]. KAN0441571C is the second generation of a ROR1 tyrosine kinase inhibitor (TKI) that induced apoptosis and death of DLBCL cells and small cell carcinoma of the lung, and that acted synergistically with venetoclax on tumor cell apoptosis [13,47].

In this study, ROR1 expression was analyzed in NSCLC tumors from patients as well as in cell lines. The anti-tumor effects of KAN0441571C alone were evaluated in vitro and in combination with erlotinib (EGFR inhibitor) and ibrutinib (BTK inhibitor). Erlotinib is a standard drug for the treatment of lung cancer [48] and ibrutinib has been suggested to be evaluated in lung cancer treatment [49,50]. These three drug molecules represent compounds with different mechanisms of action (MOA). Our data indicate that KAN0441571C in combination with EGFR or BTK inhibitors had synergistic apoptotic effects on lung cancer cells.

## 2. Materials and Methods

### 2.1. Patients

In total, 287 surgically resected NSCLC tumor specimens were analyzed for ROR1 expression. The use of patient samples was in accordance with the Declaration of Helsinki and ethically approved by the national ethics committee (www.etikprovningsmyndigheten.se) (accessed on 3 August 2015). Diagnosis was based on the WHO classification [51,52,53]. Patients were diagnosed, registered, and treated at the Oncology Department of Karolinska University Hospital Solna, Stockholm, Sweden, according to Swedish national guidelines. Based on histology, patients were divided in 3 groups including non-squamous adenocarcinoma (*n* = 157), squamous (*n* = 106), and neuroendocrine (mainly carcinoids) (*n* = 24) tumors (Appendix A). Of these, 118, 89, 40, and 9 patients were at stages IA, IB, II, and III-IV, respectively (Appendix A).

### 2.2. Immunohistochemical (IHC) Assays

Immunohistochemistry (IHC) staining of ROR1 was performed as previously described [54], using an anti-ROR1 polyclonal antibody (Proteintech, Manchester, UK). ROR1 expression was scored as negative (0), weak (1), or strong (2) according to staining intensity [6]. Slides were blindly scored by 3 independent pathologists.

### 2.3. Lung Cancer Cell Lines

Five ROR1^+^ NSCLC cell lines were used for in vitro experiments and obtained from ATCC: NCI-H1975 (EGFR mutated L858R/T790M), NCI-H23 (EGFR wild type (WT)), NCI-HCC827 (EGFR mutated E746-A750 del19), A549 (EGFR WT), and NCI-H1299 (EGFR WT) (Appendix A). 

### 2.4. ROR1, BTK and EGFR Inhibitors 

KAN0441571C, an SMI targeting ROR1, was produced and developed by Kancera AB (Stockholm, Sweden) [13,46]. Erlotinib (EGFR inhibitor) and ibrutinib (BTK inhibitor) were purchased from Selleckchem (Rungsted, Denmark).

### 2.5. Flow Cytometry

Surface ROR1 expression was evaluated by flow cytometry, as previously described [46]. Briefly, cell lines (single-cell suspension) were harvested and washed in 100 µL phosphate buffered saline (PBS) and re-suspended in 100 µL cell staining buffer (BD Biosciences, San Jose, CA, USA). Next, 10^6^ cells were incubated with an APC (allophycocyanin)-conjugated anti-ROR1 monoclonal antibody (Miltenyi Biotec, Bergisch Gladbach, Germany) and an isotype-matched antibody (Miltenyi Biotec) for twenty minutes at room temperature (RT). Cells were then washed with cell staining buffer (BD Biosciences) and were prepared for analysis by flow cytometry (Canto II, BD Biosciences). To analyze the data, the FlowJo program was used (Tree Star Inc., Ashland, OR, USA) [13].

### 2.6. Western Blot Analysis

Western blot was used to analyze the expression of proteins, as previously described [46]. Cells were lysed on ice for 1 h by lysis buffer [46]. Supernatants were collected, and a BCA Kit (ThermoFisher Scientific, Bartlett, IL, USA) was used to measure the protein concentration. Next, 10 to 20 µg of the protein lysate was mixed with loading dye and reducing agent, and then loaded onto 8 or 10% polyacrylamide gel (ThermoFisher Scientific). Proteins were separated by electrophoresis and transferred to a polyvinylidene fluoride membrane (Millipore Corporation, Bedford, MA, USA). Membranes were then blocked in 5% bovine serum albumin (BSA) (Santa Cruz Biotechnology, Heidelberg, Germany) or skimmed milk (Sigma-Aldrich, St. Louis, MO, USA). After blocking, membranes were incubated with the primary antibodies at 4 °C for overnight, washed and probed with a peroxidase (HRP)-conjugated mAb (Dako Cytomation, Glostrup, Denmark). Finally, membranes were washed and a chemiluminescence detection system (GE Healthcare, Uppsala, Sweden) was used for protein visualization. The following primary antibodies were used for protein staining: phospho (p) ROR1 (against amino acid residues Ser 652, Tyr 641, 646) [46], anti-ROR1 antibody (R&D Systems, Minneapolis, MN, USA), anti-EGFR and anti-pEGFR (Y1173) antibodies, anti-SRC and anti-p-SRC (Tyr 416) antibodies, anti-AKT and anti-p-AKT (Ser 473) antibodies, anti-mTOR and anti-p-mTOR (Ser 2448) antibodies, anti-cAMP response element-binding protein (CREB) and anti-p-CREB (Ser 133) antibodies, anti-cleaved poly ADP ribose polymerase (PARP) antibody, anti-B-cell lymphoma (BCL)-2 antibody, anti-BAX antibody, anti-myeloid cell leukemia (MCL)-1 antibody, anti-cleaved caspases 3, 8, 9 antibodies (Cell Signaling Technology, Danvers, MA, USA), anti-phosphoinositide 3-kinase (PI3K) p110δ and anti-p-PI3Kp110δ (Tyr 485) antibodies (Santa Cruz), and anti-β-actin antibody (Sigma-Aldrich). Image J 1.44p software (National Institute of Health, Bethesda, MD, USA) was used for the densitometric measurement of proteins. Ratios of phosphorylated protein/total protein were calculated.

### 2.7. MTT Cytotoxicity Assay

The MTT [3-(4,5-Dimethylthiazol-2-yl)-2,5-Diphenyltetrazolium Bromide] assay (Sigma-Aldrich) was applied to evaluate the cytotoxicity of KAN0441571C, ibrutinib, and erlotinib. Briefly, 20,000 lung cancer cells were cultured in 96 well plates in 200 µL of RPMI-1640 (ThermoFisher Scientific) (triplicates) containing 10% FBS and the drugs (diluted in DMSO). The maximum concentration of DMSO in the medium was 1%. Cells were then incubated at 37 °C for several time points (24, 48, and 72 h), followed by adding 20 µL of 5 mg/mL of the MTT solution to each well, and then further incubated for 4 h at 37 °C. Next, 100 microliters of MTT stop solution (10% SDS in 0.01 M HCL) was then added to wells and incubated for 2–4 h at 37 °C. Cells treated with DMSO alone were used as the control. Optical density (OD) was defined using a plate reader at 570 nm. The effects of the drugs in vitro on the apoptosis (additive, synergistic, or antagonist effects) of the combined treatment of KAN0441571C with erlotinib or ibrutinib was evaluated by the Chou–Talalay method using the CompuSyn software (Combosyn Inc., New York, NY, USA) [55].

### 2.8. Apoptosis Assay (Flow Cytometry)

Lung cancer cells were incubated with the drugs and cultured in 6-well plates (10^6^/well) at 37 °C for 24 h. After incubation, cells were prepared and stained for apoptosis assay as previously described [13,46]. 

### 2.9. Immunofluorescence (IF) Assay

The immunofluorescent (IF) assay was completed as previously described [46]. Briefly, cell lines were cultured on a sterile 8-well glass slide (BD Biosciences) for 24 hours to form a monolayer. Next, 4% formaldehyde was used to fix cells (15 min), which were then washed with PBS and blocked for 2 h in buffer containing 0.01% sodium azide, 2% bovine serum albumin, and 1% Tween 20 in PBS buffer. Cells were incubated with an anti-ROR1 antibody (2 µg/mL) (Sigma-Aldrich) and a non-specific antibody (mouse IgG) (eBioscience, San Diego, CA, USA) for 24 h at 4 °C, washed with PBS, and then treated with Alexa Flour 488-conjugated goat anti-mouse IgG (1:200) (ThermoFisher Scientific) for 1 h. After washing, the cell nuclei were stained with mounting media (VectaShield H-1000) containing 4’,6-diamidino-2-phenylindole (DAPI) (Vector Laboratories, Burlingame, CA, USA). A Zeiss Axioplan2 fluorescence microscope (Oberkochen, Germany) with a 63 X objective lens and the ZEN software (Carl Zeiss Microscopy, Munich, Germany) was used to take pictures.

### 2.10. Colony Formation Assay 

Two-hundred cells per well were seeded in 6 well plates (BD Biosciences) and incubated at 37 °C overnight. KAN0441571C and DMSO alone (control) were added and the cells were incubated for 72 h at 37 °C. The medium was removed and the cells were washed with PBS. Fresh medium was added and cell colonies were maintained in culture until control cultures were confluent. The medium was then removed and the colonies were stained with 4% methylene blue/MeOH (Sigma-Aldrich) for 30 min. Finally, the cells were washed with water and the plates were left to dry. Colonies were counted and the relative colony count to DMSO control was calculated using the following formula: plating efficiency (%) = number of colonies (treated)/average number of colonies (DMSO) × 100. For each cell line, the assay was repeated 2–4 times.

### 2.11. Scratch Wound Healing Assay (Migration Assay)

Lung cancer cells were seeded in 96-well plates (6 × 10^4^ cells/well) and incubated overnight at 37 °C. When cells reached >90% confluence, the IncuCyte^®^ wound maker tool (Sartorius AG, Gottingen, Germany) was used to scratch all wells uniformly. Cell migration in the presence of KAN0441571C was assessed by measuring the migration of cells into the scraped wound area. The wound-healing process was captured after 0, 24, and 48 h by IncuCyte^®^ S3 Live-Cell Analysis System (Sartorius). Three different parameters, namely wound width (μM), wound confluence (%), and relative wound density (%), were determined by scratch wound cell migration analysis using IncuCyte™ Software. For each cell line, the test was done in quadruplicate.

### 2.12. Cell Cycle Analysis

For cell cycle analysis, 100,000 A549 cells/well were seeded in 6-well plates and incubated at 37 °C overnight. KAN0441571C was added (500–5000 nM) in medium with 10% FBS and incubated for a further 24 h. DMSO 1% was used as the control. After incubation, the medium was removed and the wells were washed once with PBS. Adherent cells were detached by trypsin, transferred to tubes, and washed twice with PBS. Next, 700 μL of ice-cold 100% ethanol was added drop-wise while vortexing the tube. The tubes were incubated for 30 min to 1 h on ice and kept at 4 °C for 24 h. For propidium iodide (PI) staining, the cells were centrifuged at 1000 g at 4 °C for 10 min. The supernatant was discarded and the cells were washed with 250 μL cold PBS with 2% BSA. The samples were then centrifuged at 1000 g, at 4 °C for 10 min. The cells were re-suspended in 300 μL of staining buffer containing 40 μg/mL PI, 100 μg/mL RNase A, and 0.1% Triton X 100/PBS. The cells were transferred to FACS tubes, incubated for 20 min in RT in the dark, and then analyzed by flow cytometry (Navios, Beckman Coulter, Indianapolis, IN, USA). Data was analyzed with the Kaluza software (Navios, Beckman Coulter).

### 2.13. Statistical Methods

For comparison of ROR1 expression in the patient groups, the Kruskal–Wallis test was applied. The Mann–Whitney U test was used to check the association of variables among various groups (R version 3.3.2, The R Foundation for Statistical Computing, Vienna, Austria). The Chi-square test was utilized to find differences in the expression of ROR1. Time to progression (TTP) from diagnosis was demonstrated by the Kaplan–Meier method. Overall survival (OS) was determined from the time of diagnosis to death or last follow-up. Statistical significance was assessed by the log-rank test. For multivariable analyses, Cox regression models were used. Student’s t-test and Mann–Whitney U test were utilized for comparison of EC_50_ values (GraphPad Software, Inc., La Jolla, CA, USA). *p*-values ˂ 0.05 were considered significant. Asterisks represent *p* values of: *: 0.01 to 0.05, **: 0.001 to 0.01, ***: <0.001.

## 3. Results

### 3.1. ROR1 Expression in Subtypes of Lung Cancer and Cell Lines

ROR1 expression was significantly higher in non-squamous (136/157, 86%) and squamous (60/106, 57%) lung cancer patients compared to 5/24 (21%) patients with neuroendocrine tumors (*p* = 0.0001) (Appendix A). No association between ROR1 expression and pathological disease stage was noted, irrespective of the histopathological subtype (Appendix A). In adenocarcinoma, no statistical association between ROR1 expression and age, grading, smoking status, and overall survival was seen (Appendix A). However, a trend to shorter TTP in ROR1^+^ vs ROR1^-^ patients was found (*p* = 0.09) (Appendix A). In addition, a significant higher proportion of p53 negative cases (63%) was seen in the ROR1^+^ group as compared to p53 positive patients (25%) (*p* = 0.03) (Appendix A).

In patients with squamous NSCLC, no statistically significant association was observed between ROR1 expression and age, stage, grading, smoking status, p53 expression, and overall survival.

Surface ROR1 expression was detected in all cell lines with the exception for NCI-H23 (Figure 1A). However, using Western blot and IF, ROR1 was shown to be expressed in all cell lines (Figure 1B and Appendix A). ROR1 was phosphorylated in all cell lines (Figure 1B).

### 3.2. Cytotoxicity of KAN0441571C, Ibrutinib and Erlotinib in Lung Cancer Cell Lines

KAN0441571C induced a dose-dependent cytotoxicity (MTT) in all cell lines with the exception for A549. The cytotoxic effect of KAN0441571C varied between cell lines. In comparison to ibrutinib and erlotinib, the cytotoxic effect of KAN0441571C was similar or significantly better. Both erlotinib and ibrutinib induced the highest cytotoxic effect in HCC827 with EGFR Del19 (mutated E746-A750) compared to cell lines without EGFR Del19 or of the EGFR wild-type. However, in the HCC827 cell line, a dose-dependency within the used concentrations of erlotinib and ibrutinib could not been seen (compared to KAN0441571C in A549) (Figure 2). EC_50_ values for erlotinib and ibrutinib were >10,000 nM for all cell lines with the exception for NCI-HCC827, where EC_50_ was <1250 nM for both erlotinib and ibrutinib (Figure 2).

Apoptosis was confirmed by Annexin V/PI staining (Figure 3A). A dose-response relationship was noted for all cell lines with the exception for NCI-HCC827 (compared with MTT for KAN0441571C, which seemed to be the most resistant in the Annexin V/PI assay). EC_50_ values are also shown in Figure 3A. Apoptosis was accompanied by downregulation of BCL-2, and MCL-1, as well as by cleavage of PARP and caspases 3, 8, and 9, while the BAX protein was upregulated (Figure 3B,C). Caspase 8 and 9 cleavage indicates activation of both the extrinsic and intrinsic apoptotic pathways. 

### 3.3. Colony Formation Assay

In a colony tumor cell formation assay (72 h), KAN0441571C reduced the ability of A549, NCI-HCC827, NCI-H23, and NCI-H1299 cells at concentrations ranging from 100 to 200 nM to form colonies, indicating that the drug compromised the clonogenic capacity of the cells (Figure 4) (NCI-H1975 was not included in this assay). Colonies were completely abolished at 200 nM of KAN0441571C in all cell lines. In H1299, only few colonies were seen at 125 nM. These findings indicate that inhibition of ROR1 may irreversible prevent NSCLC cells to recover after exposure to ROR1 inhibition.

### 3.4. Migration of Lung Cancer Cells

The migratory ability of lung cancer cells was analyzed by the scratch wound healing assay (Figure 5 and Appendix A). After 48 h of incubation with KAN0441571C, untreated (DMSO alone) cells proliferated and migrated as expected. KAN0441571C inhibited migration in a dose- and time-dependent manner. At 1 µM, tumor cell mobility was completely abrogated.

### 3.5. Cell Cycle Analysis

The cell cycle profile of the lung cancer cell line A549 cell was analyzed after treatment with KAN0441571C. A decrease in S phase cells, including the accumulation of G2/M phase cells and augmented cell death (sub-G1 cells), was noted with increasing concentrations of KAN0441571C in 24 h cell culture. A decrease in G1 cells was also seen at the highest concentration of KAN0441571C. Data indicate that replicating cells might be vulnerable to cell death at exposure to the ROR1 inhibitor KAN0441571C and/or that KAN0441571C induced a cytostatic effect in lung cancer cells preventing exit from G2/M phase and cell proliferation (Appendix A).

### 3.6. Effect of KAN0441571C on Signaling Molecules 

As expected [56], KAN0441571C induced dephosphorylation of ROR1 in a dose-dependent manner, as well as EGFR, SRC, PI3K110δ/AKT/mTOR (non-canonical Wnt pathway), and CREB in the NCI-H23 lung cancer cell line (Appendix A).

### 3.7. Effects on Tumor Cell Death of KAN0441571C Alone and in Combination with Ibrutinib or Erlotinib 

KAN0441571C induced a significant cell death (MTT) in all cell lines, at a concentration of 250 nM. For erlotinib and ibrutinib, significantly higher concentrations were required) 5000–10,000 nM) (Figure 6). The cytotoxic effect of KAN0441571C was significantly higher than that of erlotinib and ibrutinib with the exception of NCI-HCC827 (EGFR mutated E746-A750, Del19). When KAN0441571C was combined with ibrutinib or erlotinib, cytotoxicity increased significantly in 3 out of 5 cell lines.

To further assess the efficacy (antagonism, addition, or synergism) of the combinations of KAN0441571C/erlotinib and KAN0441571C/ibrutinib on apoptosis, the Chou–Talalay method was applied. The combinations had synergistic or additive apoptotic effect in all tested lung cancer cell lines (Figure 7 and Appendix A).

## 4. Discussion

In the present study, we analyzed ROR1 expression in primary tumors of patients with surgically resected NSCLC. In non-squamous and squamous carcinomas, 86% and 56%, respectively, expressed ROR1, while in neuroendocrine tumors only about 20% exhibited ROR1. No relationship between ROR1 expression and age, stage, tumor grading, smoking status, and overall survival was observed in adenocarcinoma or squamous NSCLC patients. However, an inverse relationship to p53 expression in non-squamous NSCLC (*p* = 0.03) was seen. Such a finding might be expected as low or no p53 expression is associated with a poor outcome in malignancies [57]. There was also a trend towards shorter TTP in ROR1^+^ adenocarcinoma NSCLC patients compared to ROR1^−^ patients.

KAN0441571C prevented ROR1 phosphorylation and inactivated both Wnt non-canonical pathway molecules and the transcription factor CREB. Apoptosis was induced in NSCLC cell lines both through the extrinsic and intrinsic pathways, including downregulation of pro-survival molecules (BCL-2 and MCL-1) and upregulation of the pro-apoptotic BAX protein. The ROR1 inhibitor was more effective than an EGFR inhibitor in inducing apoptosis. There seemed to be no relation between the response to KAN0441571C and mutations of EGFR, K-Ras, or p53. The combination of KAN0441571C with erlotinib or ibrutinib had synergistic apoptotic effects.

ROR1 is dysregulated during tumorigenesis and is constitutively expressed and phosphorylated in various tumors [13,29,31,39,58]. ROR1 has been shown to be a prognostic marker in several tumors [59]. A high level of ROR1 expression was noted to be associated with an aggressive and poor prognosis disease in, for example, CLL, DLBCL, MCL, ALL, as well as in ovarian, breast, pancreatic, gastric, colorectal, and lung cancers [15,18,21,32,59,60,61,62,63]. In meta-analyses, high ROR1 expression was found to relate to worse overall survival in hematologic malignancies and solid tumors [59,64]. The data may support an important role of ROR1 in cancer. 

ROR1 is involved in a lot of functions in tumor cells, such as proliferation, survival, migration, stemness, epithelial to mesenchymal transition (EMT), chemotaxis, and drug resistance, through the planar cell polarity (PCP) activation and Ca2^+^ dependent pathways of non-canonical Wnt signaling [25,26,65,66]. Binding of Wnt5a to ROR1 stimulates several signaling pathways, such as PI3K/AKT and RhoA/Rac1 GTPases, activating the transcriptional coactivator YAP/TAZ or polycomb complex protein BMI-1 to sustain stemness, metastatic ability, and drug resistance [66]. 

ROR1 has been described as an oncogenic molecule of interest for the development of targeted therapy for antibody-drug conjugates [67], chimeric antigen receptor T-cells (CAR-T) [68,69], mAbs [27], and bi-specific T-cell engager (BiTE) [70], as well as SMIs targeting the intracellular [20,56] or extracellular parts of ROR1 [42,43].

Targeting ROR1 by small molecules and mAbs has been shown to be an effective therapeutic approach in pre-clinical and clinical studies in various malignancies, such as CLL, MCL, lung, breast cancers, etc. [27,43,46,67,71]. The silencing of ROR1 significantly inhibited the proliferation of tumor cells in lung adenocarcinoma via the PI3K/AKT/mTOR signaling pathway [72,73]. NSCLC is still a major therapeutic challenge and there is a great need for new treatment alternatives.

In previous reports, we have described the effects of two small-molecule ROR1 inhibitors (KAN0439834 and KAN0441571C) targeting the cytoplasmic tyrosine kinase domain that dephosphorylated ROR1 and induced tumor cell apoptosis [13,20,46,74]. In the current study, we analyzed the effects of KAN0441571C in lung cancer cells. We also compared its tumor cell killing effect with erlotinib, which is one of the current drug options for the treatment of NSCLC patients with EGFR exon 19 deletions or exon 21 L858R substitution mutational status [75]. KAN0441571C was more effective than erlotinib in inducing apoptosis of lung cancer cells, irrespective of EGFR mutations. The combination of the two inhibitors had significantly improved the synergistic apoptotic effects, in the NCI-H1299, NCI-H1975, and NCI-HCC827 cell lines.

NKX2-1, a lineage-specific transcription factor, has been noted to be essential for the development of peripheral parts of the lung and in morphogenesis, and it is overexpressed in lung adenocarcinomas [24,76,77]. NKX2-1 induced ROR1 transcription and was critically involved in the maintenance of a balance between the pro-apoptotic p38 pathway and the pro-survival PI3K/AKT signaling pathway [16]. Furthermore, activated ROR1 binds to SRC and activates it. SRC activation inhibits PTEN activity and results in AKT activation. An association between EGFR mutations and NKX2-1 expression has been described in lung adenocarcinoma [78]. ROR1 has also been shown to form heterodimers with EGFR, promoting the maintenance of lung cancer cell survival [16,79]. High ROR1 expression in EGFR T790M NSCLC patients was related to an inferior progression-free survival in erlotinib-treated patients compared to those with low ROR1 expression, substantiating a role of ROR1 in the disease pathobiology and supporting that the inhibition of ROR1 might add to the therapeutic effect [79]. Overall, the data indicate a role of ROR1 expression in lung adenocarcinoma, which may be independent of ROR1 kinase activity [16].

ROR1 inhibition by KAN0441571C blocked the phosphorylation of ROR1 and EGFR. Dephosphorylation of EGFR may be due to the inhibition of the phosphorylation of ROR1, resulting in the inactivation of the ROR1-EGFR dimer and the suppression of survival signals [16,80]. 

In addition, to target the TK domain of ROR1 by KAN0441571C, ROR1 targeting SMIs against other extracellular and intracellular regions of ROR1 have been produced [42,43,46]. ARI-1 ((R)-5,7-bis (methoxymethoxy)-2-(4-methoxyphenyl) chroman-4-one) is a novel ROR1 inhibitor targeting the external CRD domain of ROR1, preventing NSCLC cell proliferation, migration, as well as inactivating the PI3K/AKT/mTOR signaling pathway [42]. ARI-1 decreased the growth of ROR1-expressing NSCLC in vitro and in vivo with no significant side effects in mice [42]. This inhibitor seemed to interrupt Wnt5a binding to ROR1 via CRD domain blockage. 

Strictinin, isolated from *Myrothamnus flabellifolius*, targeted the intracellular region of ROR1 and induced apoptosis of triple-negative breast cancer cells (TNBC) through the intrinsic apoptotic pathway. Therefore, this inhibitor prevented ROR1 activation via inhibiting the binding to Wnt5a, dephosphorylating AKT and GSK3β, and blocking ROR1 signaling in a β-catenin-independent PI3K/AKT/GSK3ß signaling pathway [43].

In contrast to AR-1 and strictinin, KAN0441571C may not directly inhibit the binding of Wnt5a to ROR1, but instead might change the conformational structure of ROR1 [42,43], which thus prevents ROR1 from binding with Wnt5a. The treatment of human cells with KAN0441571C has also been shown to dephosphorylate SRC, which binds to phosphorylated intracellular regions of ROR1 [81].

In conclusion, ROR1 is expressed in NSCLC and seems to be of importance in the pathobiology of the disease. A ROR1 small molecule inhibitor (KAN0441571C) was highly effective in inducing apoptosis of NSCLC cells, and was superior to erlotinib and ibrutinib. The combination of KAN0441571C with erlotinib had significant synergistic cytotoxic effects. The development of new targeted drugs with other MOAs than those clinically available is warranted to improve the prognosis in NSCLC patients. A ROR1 inhibitor may be a new promising drug candidate.

## Figures and Tables

**Figure 1 pharmaceutics-15-01148-f001:**
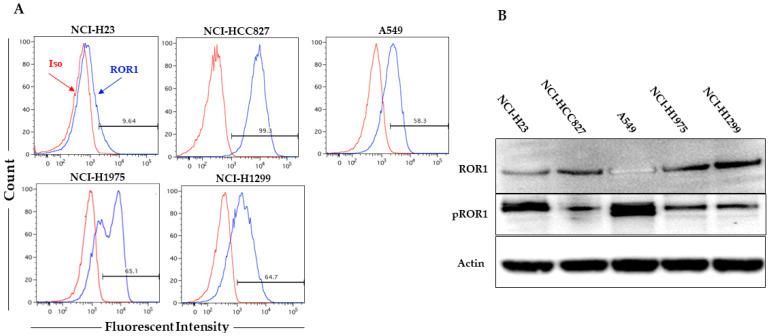
(**A**) Surface staining for ROR1 (flow cytometry) (red line: isotype control; blue line: anti-ROR1 antibody); (**B**) Western blot for ROR1 and pROR1 expression (130 KDa) in 5 lung cancer cell lines. Actin was used as internal control.

**Figure 2 pharmaceutics-15-01148-f002:**
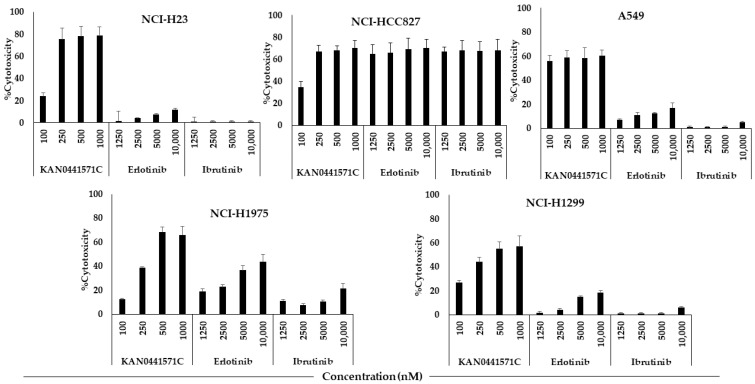
Cytotoxicity (MTT) (72 h) in 5 lung cancer cell lines incubated with various concentrations of KAN0441571C, erlotinib, or ibrutinib.

**Figure 3 pharmaceutics-15-01148-f003:**
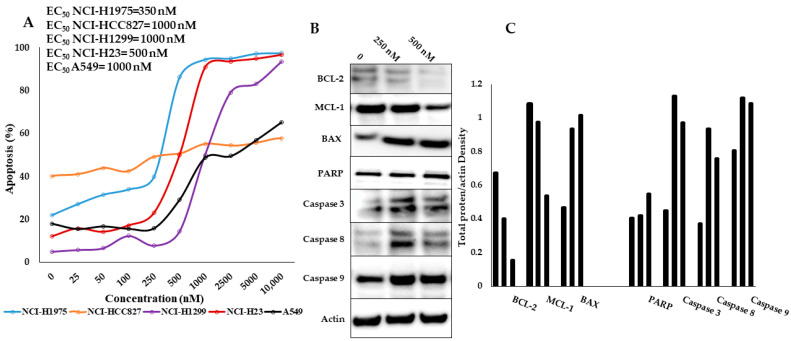
(**A**) Apoptosis (Annexin V/PI) of KAN0441571C in 5 lung cancer cell lines (EC_50_ values are shown to the upper left). (**B**) Western blots of apoptotic proteins BCL-2, MCL-1, BAX, PARP, and caspases-3, 8, and 9 after 24 h of incubation with KAN0441571C in NCI-H23 cells. One representative data of three individual experiments are shown. (**C**) Densitometric measurements of protein bands in Figure 3B expressed as total protein/actin intensity.

**Figure 4 pharmaceutics-15-01148-f004:**
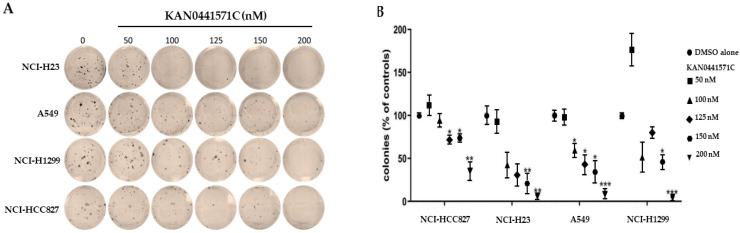
(**A**) Lung cancer cell lines (NCI-H23, A549, NCI-H1299, and NCI- HCC827) incubated with KAN0441571C (72 h) prevented formation of cell colonies in a dose-dependent manner (representative images). (**B**) Relative quantification of colonies (mean ± SEM) at various concentrations of KAN0441571C. * *p* < 0.05, ** *p* < 0.01, *** *p* < 0.001 (one-way ANOVA).

**Figure 5 pharmaceutics-15-01148-f005:**
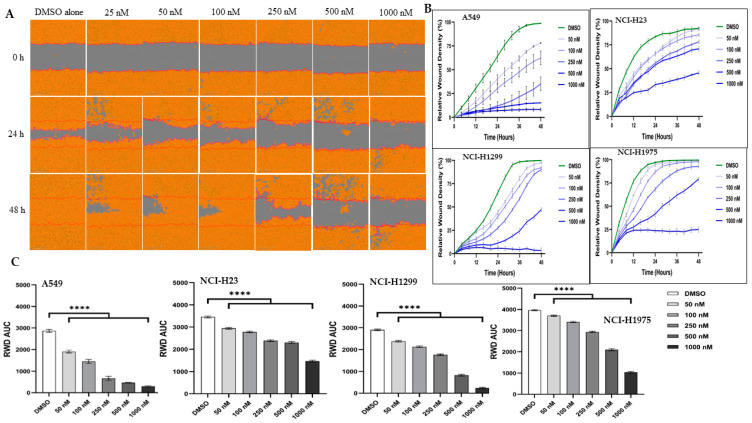
KAN0441571C inhibited migration of lung cancer cell lines in the scratch wound healing assay. (**A**) Migration of A549 was inhibited as shown by images acquired after 0, 24, and 48 h of incubation with different concentrations of KAN0441571C. Images of 1 representative experiment out of 4 for each cell line is shown. (**B**) Rate of wound closure (relative wound density) of lung cancer cells. The Incucyte Zoom Imager Software Analyzer was used to count the number of lung cancer cells during 48 h of incubation with of KAN0441571C (25–1000 nM). DMSO used as control. (**C**) Reduction (mean ± SEM) of cell migration (RWD AUC) after treatment with KAN0441571C (0–1000 nM) in 4 cell lines. **** *p* < 0.0001.

**Figure 6 pharmaceutics-15-01148-f006:**
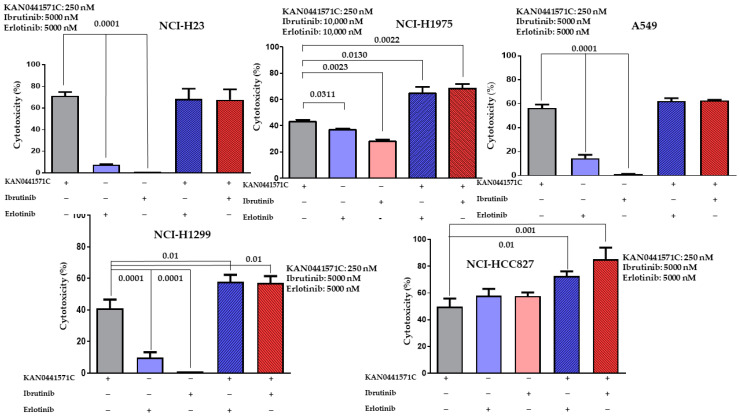
Cytotoxicity (MTT) (mean ± SEM) of five lung cancer cell lines incubated with KAN0441571C in combination with ibrutinib or erlotinib. Results of 3 independent experiments are shown. The concentration of the respective drug are shown for each cell line (−: No inhibitor added, +: inhibitor added).

**Figure 7 pharmaceutics-15-01148-f007:**
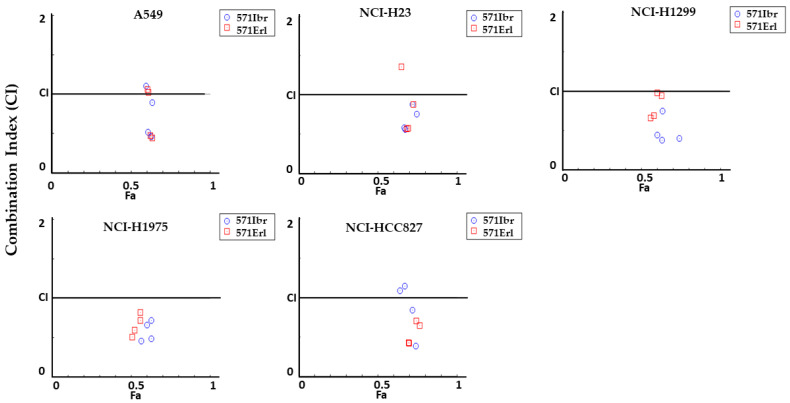
Combination index (CI) plots for apoptosis (Annexin V/PI) in lung cancer cell lines incubated (24 h) with a combination of KAN0441571C (250 nM) with ibrutinib (571Ibr) (blue symbols) or erlotinib (571Erl) (red symbols) (5000–10,000 nM, see Figure 6). Values across Fa in lung cancer cells are shown, where Fa is the % of cell death. CI < 1 synergistic effect; CI = 1 additive effect; CI > 1 antagonistic effect (Chou–Talalay method).

## Data Availability

Not applicable.

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
