# Peer review of "A Small Molecule Targeting the Intracellular Tyrosine Kinase Domain of ROR1 (KAN0441571C) Induced Significant Apoptosis of Non-Small Cell Lung Cancer (NSCLC) Cells"

_pharmaceutics, 2023, doi:10.3390/pharmaceutics15041148_

Round 1

Reviewer 1 Report

The authors investigated the expression of ROR1 in non-small cell lung cancers and the correlation with disease stage, P53 status, and progression. Then used in vitro methods of testing the KAN0441571C small molecule as ROR1 inhibitor on five lung cancer cells (that have wild type or mutated EGFR) they used apoptosis and cytotoxicity assay to demonstrate that KAN0441571C is more effective in cell killing than EGFR inhibitors (erlotinib and ibrutinib) in addition to the synergistic effect when used in combination.

Methods

In line 114 (ROR1 expression was scored as negative (0), weak (1) or strong (2) according to staining intensity) was the scoring performed blindly by another investigator?

In line 119 (Five ROR1+ NSCLC cell lines were used for in vitro experiments and obtained from ATCC), did the investigator also use ROR1- NSCLS or only the ROR1 positive?

In line 132 (cell staining buffer), if this is commercially obtained, please mention the source otherwise, state the composition.

In line 189 (Lung cancer cells were incubated with the drugs and cultured in 6-well plates (106/well)), please correct to 106 /well.

Results

In figure 1A in line 289, the flow cytometry panel is cut, so the x-axis scales (numbers) are not shown, which makes it difficult to confirm if the analysis was performed using the log scale

In line 304 author stated, “EC50 values for erlotinib and 304 ibrutinib was >10000 nM for all cell lines with the exception for NCI-HCC827 305 where EC50 was <1250 nM for both erlotinib and ibrutinib” and refers to figure 2. Figure 2 shows the maximum dose tested as 10000

In line 307, figure 2, why are there no error bars?

In line 321, figure 3, the author used NCI-H23 cell line as a model, and figure 3A shows that KAN0441571C dose of 250 cause apoptosis in 20%, and this jumps to 50% at 500nM; in previous figure 2, the cytotoxicity of the two doses seemed similar, is there an explanation for such discrepancy?

It will be interesting if the authors knock down ROR1 and investigate if this will abolish the effect of the tested molecule (KAN0441571C). 

Ageneral note: the sample size (n) is missing from most of the figure legends or results section.

Discussion:

it would be of benefit that the author address ROR1 investigated in lung cancers by other groups; there are multiple publications that investigated ROR1 in lung cancers.

In line 467 (KAN0441571C was more effective than erlotinib in inducing apoptosis of lung cancer cells, irrespective of EGFR mutations. The combination of the two inhibitors had synergistic apoptotic effects). This statement is not correct as not all the tested cells showed synergistic effects like H23 and A549.

In line 505 (Our ROR1 small molecule inhibitor might also prevent binding of intracellular adaptor molecules as SRC to ROR1 interrupting survival signaling cascades), the author did not investigate such claim experimentally, IP-experiment needed to show the biding is abolished or reduced after treatment with KAN0441571C.

Author Response

Reviewer 1

Comments and Suggestions for Authors

The authors investigated the expression of ROR1 in non-small cell lung cancers and the correlation with disease stage, P53 status, and progression. Then used in vitro methods of testing the KAN0441571C small molecule as ROR1 inhibitor on five lung cancer cells (that have wild type or mutated EGFR) they used apoptosis and cytotoxicity assay to demonstrate that KAN0441571C is more effective in cell killing than EGFR inhibitors (erlotinib and ibrutinib) in addition to the synergistic effect when used in combination.

Dear Reviewer

Many thanks for the excellent comments increasing the value of our manuscript.  We here respond to your comments one by one.

Methods

  • In line 114 (ROR1 expression was scored as negative (0), weak (1) or strong (2) according to staining intensity) was the scoring performed blindly by another investigator?

Response: Slides from each patient were evaluated and scored blindly by 3 pathologists. A statement has been added (line 116).

  • In line 119 (Five ROR1+ NSCLC cell lines were used for in vitro experiments and obtained from ATCC), did the investigator also use ROR1- NSCLC or only the ROR1 positive?

Response: NCI-H23 is negative for ROR1 surface expression (flow-cytometry), but express ROR1 WB of lysate. We have analysed several NSCLC cell lines, but we could not find a cell line that did not express ROR1 as detected by WB of cell lysate.

In our previously published paper in DLBCL (PMID: 32586008, present, revised Ref. No. 13), we found a ROR1 negative cell line (U2932). Our ROR1 small molecule inhibitor (KAN0441571C) had no significant effect on cell death of this cell line (EC50>2000 nM).   

  • In line 132 (cell staining buffer), if this is commercially obtained, please mention the source otherwise, state the composition.

Response: Included.

  • In line 189 (Lung cancer cells were incubated with the drugs and cultured in 6-well plates (106/well)), please correct to 10/well.

Response: Done.

Results

  • In Figure 1A in line 289, the flow cytometry panel is cut, so the x-axis scales (numbers) are not shown, which makes it difficult to confirm if the analysis was performed using the log scale.

Response: Corrected.

  • In line 304 author stated, “EC50values for erlotinib and 304 ibrutinib was >10000 nM for all cell lines with the exception for NCI-HCC827 305 where EC50 was <1250 nM for both erlotinib and ibrutinib” and refers to Figure 2. Figure 2 shows the maximum dose tested as 1000.

Response: For all cell lines, we also tested 25000 nM for erlotinib as well as for ibrutinib but EC50 values were not reached.

  • In line 307, Figure 2, why are there no error bars?

 Response: Corrected.

  • In line 321, Figure 3, the author used NCI-H23 cell line as a model, and Figure 3A shows that KAN0441571C dose of 250 cause apoptosis in 20%, and this jumps to 50% at 500nM; in previous Figure 2, the cytotoxicity of the two doses seemed similar, is there an explanation for such discrepancy?

Response: In Figure 2 we showed MTT cytotoxicity data and in Figure 3 AnnexinV /PI (apoptosis) flow cytometry results. MTT values are usually higher as it is based on drug cytotoxicity. Due to differences in these two assays, you might expect to see such variations.

  • It will be interesting if the authors knock down ROR1 and investigate if this will abolish the effect of the tested molecule (KAN0441571C).

Response: Many thanks for your valuable comment. We will consider this experiment in the future.

  • A general note: the sample size (n) is missing from most of the Figure legends or results section.

 Response: Added.

Discussion:

  • it would be of benefit that the author address ROR1 investigated in lung cancers by other groups; there are multiple publications that investigated ROR1 in lung cancers.

Response:  We added and referred to appropriate references of other studies in lung cancer.

  • In line 467 (KAN0441571C was more effective than erlotinib in inducing apoptosis of lung cancer cells, irrespective of EGFR mutations. The combination of the two inhibitors had synergistic apoptotic effects). This statement is not correct as not all the tested cells showed synergistic effects like H23 and A549.

Response: The statement has been corrected accordingly now.

  • In line 505 (Our ROR1 small molecule inhibitor might also prevent binding of intracellular adaptor molecules as SRC to ROR1 interrupting survival signaling cascades), the author did not investigate such claim experimentally, IP-experiment needed to show the biding is abolished or reduced after treatment with KAN0441571C.

Response:  The statement has been deleted.

Reviewer 2 Report

the authors look at the effect of KAN0441571C on NSCLC cell lines. The work is very interesting but there are some things that could be improved:

- This paper focuses on cell lines basically so I would remove the patient part. So in the abstract there are things that I would not put as line 26-29. 

- Why didn't they do the same with primary culture and not with cell lines?

- In the methodology there are missing cities for example line 149, 153 and 165.

- First time an abbreviation appears, put it in. Example PBS, line 132.

- Line 110-112, the consent part should be in the patient part.

- Figure 1B, the houskeeping is missing in the western.

- In the figures, if there is any significant data, it has not been highlighted. 

- Item 3.7 I would put it next to 3.2 as they are both cytotoxicity.

- Figure 3C shows proten and not protein. In addition, it seems that the columns are cut.

- Why didn't they use the NCI-H1975 for the column formation experiment?

Author Response

Reviewer 2

Comments and Suggestions for Authors

the authors look at the effect of KAN0441571C on NSCLC cell lines. The work is very interesting but there are some things that could be improved:

Dear Reviewer,

 Many thanks for your excellent comments that increased the value of our manuscript. Here we respond to your comments one by one.

  • This paper focuses on cell lines basically so I would remove the patient part. So, in the abstract there are things that I would not put as line 26-29. 

Response: We started this study by testing the ROR1 expression in lung cancer tissues from NSCLC patients. We continued our experiments on cell lines as an available model for lung cancer cells. We prefer to keep this part of the study as the information in this regard in the literature is limited.

  • Why didn't they do the same with primary culture and not with cell lines?

Response:  It is very difficult for cancers of epithelial origin to establish tumor cell cultures and from excised tumor specimens and expand such cell lines. Moreover, most of the surgically removed tumor is used for diagnostic purposes and biobanking. With these restrictions, we decided to analyse the effects of the ROR1 small molecule inhibitor on commercially available cell lines.

  • In the methodology there are missing cities for example line 149, 153 and 165.

 Response: Completions have been done.

  • First time an abbreviation appears, put it in. Example PBS, line 132.

Response: Done.

  • Line 110-112, the consent part should be in the patient part.

Response:  Done.

  • Figure 1B, the housekeeping is missing in the western.

 Response: To check the phosphorylation status of a molecule, normally the total protein is considered as loading control.

  • In the Figures, if there is any significant data, it has not been highlighted. 

 Response: To reduce the length of the manuscript, we have highlighted significant data.

  • Item 3.7 I would put it next to 3.2 as they are both cytotoxicity.

Response: We started with presenting single agent treatment data and then continued with combination experiments. For us this seems to be most logic way.

  • Figure 3C shows protein and not protein. In addition, it seems that the columns are cut.

 Response: In Figure 3C we measured the density of main band of pro-apoptotic/pro-survival protein bands to actin as an internal control. Figure 3C is a confirmation of Figure 3B.

We have uploaded the full blots of all cut blots. Full blots should be available on the webpage.

  • Why didn't they use the NCI-H1975 for the column formation experiment?

Response: We had  technical problems when testing this specific cell line in CFA and could not get representative pictures.

Reviewer 3 Report

The effects of ROR1 inhibitor KAN0441571C in NSCLC cells were investigated in the manuscript.

1. This manuscript is submitted to the ‘Biologics and Biosimilars’ section. This manuscript is about small-molecule drugs, not biologics. Authors and editors need to change the SECTION.

2. Figure 2 should be re-evaluated with a broader dose range for a more reliable GI50 calculation.

3. Authors should explore the possibility of off-target effects of KAN0441571C. Given the similarity of ROR1 and Trk and Trk is the current NSCLC target, Trk inhibition by KAN0441571C should be checked.

4. Line 130: flowcytometry -> flow cytometry

5. Line 198: immunofluorescent assay -> immunofluorescence assay

Author Response

Reviewer 3

Comments and Suggestions for Authors

The effects of ROR1 inhibitor KAN0441571C in NSCLC cells were investigated in the manuscript.

Dear Reviewer,

 Many thanks for your excellent comments which increased the value of our manuscript. Here we respond to your comments one by one.

  • This manuscript is submitted to the ‘Biologics and Biosimilars’ section. This manuscript is about small-molecule drugs, not biologics. Authors and editors need to change the SECTION.

Response: We will ask the Editor-in-Chief to make the necessary change.

  • Figure 2 should be re-evaluated with a broader dose range for a more reliable GI50 calculation.

Response: We have tested the effects of the small molecules at higher concentrations in all cell lines. However, the EC50 was not reached even at higher concentrations.

  • Authors should explore the possibility of off-target effects of KAN0441571C. Given the similarity of ROR1 and Trk and Trk is the current NSCLC target, Trk inhibition by KAN0441571C should be checked.

 Response: We have previously published off-target effects of the 1st generation of our ROR1 inhibitor (KAN0439834). KAN0439834 is similar to KAN0441571C ((Leukemia. 2018 Oct;32 (10): 2291-2295. doi: 10.1038/s41375-018-0113-1. PMID: 29725030) (Supplementary Methods, Information and Figures) (Ref. No. 46 in the present, revised manuscript). The off targets were defined by the KINOMEscan™ (DiscoverX, San Diego, CA, USA) assay. In this profiling, a competition assay was used to measure binding interactions between the test compound (KAN0439834) and 456 human kinases and disease relevant mutant variants (395 non-mutant kinases). Furthermore, using a radiometric assay (ProQinase) (PMID: 32586008, Ref. No. 13 in present, revised version, Table S3) we compared KAN0439834 and KAN0441571C.  No off-target effect was observed for the Trk family members.

  • Line 130: flowcytometry -> flow cytometry

 Response: Done.

  • Line 198: immunofluorescent assay -> immunofluorescence assay

 Response: Done.

Round 2

Reviewer 2 Report

 Thanks to the authors for answering my questions. The manuscript is very interesting but there is still some room for improvement.

- line 168 does not read well, maybe it is because of the proofreading.

- In the methodology the cities are still missing, line 176,178.... revise this whole section.

- Line 195, I suppose that PI is Propidium iodide, but put what it is.

- In figure 1B if they use total protein how was it quantified? Where is the image?

- In figure 3C ok it is confirmation of 3B but the axis has to be larger as some seem to be cut off for example caspase 9 with 250-500nM.

Author Response

Reviewer 2

Comments and Suggestions for Authors

 Thanks to the authors for answering my questions. The manuscript is very interesting but there is still some room for improvement.

  • line 168 does not read well, maybe it is because of the proofreading.

Response: Yes. It is because of proofreading. We will correct during the proof-reading process.

2- In the methodology the cities are still missing, line 176,178.... revise this whole section.

Response:  We have corrected as recommended per the journal instruction. If we have to repeat the name of the provider more than once, we only use the name of company.

3- Line 195, I suppose that PI is Propidium iodide, but put what it is.

Response:  Corrected.

4- In figure 1B if they use total protein how was it quantified? Where is the image?

Response: we used actin staining as internal control (included). We did not measure the density of bands as we only want to show that ROR1 and phospho-ROR1 are expressed in the cell lines.

5- In figure 3C ok it is confirmation of 3B but the axis has to be larger as some seem to be cut off for example caspase 9 with 250-500nM.

Response: We set 1 as the upper limit for Y axis.  We have replaced Fig 3C to the original one.

Reviewer 3 Report

Manuscript contents themselves are appropriate for publication in Pharmaceutics.

However, the manuscript are submitted to the wrong section, Biologics and Biosimilars.

The section should be corrected before the acceptance.

Author Response

Reviewer 3

Comments and Suggestions for Authors

Manuscript contents themselves are appropriate for publication in Pharmaceutics.

However, the manuscript is submitted to the wrong section, Biologics and Biosimilars.

The section should be corrected before the acceptance.

Response:  The editorial Board allocated this manuscript to a specific issue i.e. to the Section ``Biologics and Biosimilars`` in  a special issue: ``Kinase Inhibitor for Cancer Therapy``.  After acceptance of the paper, the editorial board will decide about the allocation. We have asked the editorial board to check your comment.